# Kidney Function, Male Gender, and Aneurysm Diameter Are Predictors of Acute Kidney Injury in Patients with Abdominal Aortic Aneurysms Treated Endovascularly

**DOI:** 10.3390/toxins15020130

**Published:** 2023-02-04

**Authors:** Bartłomiej Antoń, Sławomir Nazarewski, Jolanta Małyszko

**Affiliations:** 1Department of General, Vascular and Transplant Surgery, Medical University of Warsaw, 02-097 Warsaw, Poland; 2Department of Nephrology, Dialysis and Internal Medicine, Medical University of Warsaw, 02-097 Warsaw, Poland

**Keywords:** abdominal aortic aneurysm, acute kidney injury, EVAR

## Abstract

Abdominal aortic aneurysm (AAA) is a degenerative disease of the aortic wall with potentially fatal complications. The widespread adoption of endovascular aneurysm repair (EVAR), which is less invasive and equally (if not more) effective for abdominal aortic aneurysms (AAA), is due to the obvious advantages of the procedure compared to the traditional open repair. As the popularity of endovascular procedures grows, related complications become more evident, with kidney damage being one of them. Although acute kidney injury following EVAR is relatively common, its true incidence is still uncertain. The purpose of this study was to assess the incidence of acute kidney injury among patients treated with endovascular repair of ruptured AAA. In addition, we aimed to determine the predictors of PC-AKI in patients with abdominal aortic aneurysm treated with EVAR. Patients and Methods: We retrospectively analyzed a prospective registry of abdominal aortic aneurysm of 247 patients operated endovascularly at a single center between 2015 and 2021. Due to a lack of clinical data, data of 192 patients were reviewed for postcontrast acute kidney injury. Additional comorbidities were included in this study: hypertension, diabetes mellitus, atrial fibrillation, chronic coronary syndrome, COPD, and chronic kidney disease. Follow-up examinations were performed before the procedure and 48 h after contrast administration. Results: The group of 36 patients developed PC-AKI, which is 19% of the entire study population. Hypertension, diabetes, chronic kidney disease, male gender, and incidence of PC-AKI were more prevalent in patients with higher aortic aneurysm diameter ≥67 mm. In multiple regression analyses, independent predictors of PC-AKI were serum creatinine, chronic kidney disease, male gender, and aortic aneurysm diameter ≥67 mm. Conclusions: One of the major complications after EVAR is acute kidney injury, which is linked to higher death and morbidity rates. Independent risk factors for postcontrast acute kidney injury were chronic kidney disease, male gender, and aortic diameter. Only aortic diameter could be modifiable risk factor, and earlier surgery could be considered to yield better outcomes. More research is critically needed to determine how AKI affects long-term outcomes and to look at preventive options.

## 1. Introduction

In clinical medicine, contrast agents are frequently used to enhance the visibility of interior organs and structures in X-ray-based imaging procedures including computed tomography (CT) and radiography. High contrast density is achieved by iodine contrast media, which contain one or two tri-iodobenzene rings. Iodine-based contrast agents are often categorized into iso-, low-, and high-osmolar substances based on their osmolarity. In recent years, the use of contrast agents has increased, particularly for several diagnostic imaging procedures such as angiography [1]. From minor symptoms, such as itching, to major and even fatal reactions, such anaphylactic reactions, radiographic contrast media can cause a variety of adverse reactions. Kidney failure is known to be one of the main side consequences of administering contrast media. When contrast medium is administered intravenously or intraarterially, a serious response known as contrast-induced nephropathy (CIN) can occur. CIN is defined as an acute renal failure with no other cause that results in a serum creatinine rise of at least 25% (or 44 mol/L or 0.5 mg/dL) following the administration of contrast agents [2]. The criteria used for postcontrast acute kidney injury (PC-AKI) research, developed by the Kidney Disease Improving Global Outcome (KDIGO) initiative, are more accurately derived than those utilized by the CIN definition. KDIGO defines PC-AKI as acute kidney injury caused secondary to contrast administration. The definition of AKI provides one of the following: an increase in SCr of ≥0.3 mg/dL (≥26.5 µmol/L) within 48 h; an increase in SCr of ≥1.5 times baseline that is known or presumed to have occurred within the previous 7 days; or an increase in urine volume of 0.5 mL/kg/h for 6 h. AKI can be divided into three stages. Stage one requires one of the following: a reduction in urine production to <0.5 mL/kg/h during a 6 h block, increase in SCr by ≥0.3 mg/dL (≥26.5 µmol/L) within 48 h, or increase in SCr to ≥1.5–1.9 times baseline, which is known or presumed to have occurred within the prior 7 days. Stage two demands one of the following: serum creatinine increase ≥ 2.0–2.9 times baseline or reduction in urine production to <0.5 mL/kg/h during two 6 h blocks. Stage three finally is defined by one of the following: serum creatinine increase ≥3.0 times baseline, serum creatinine increase to >4.0 mg/dL (353 µmol/L), initiation of renal replacement therapy, reduction in urine production to <0.3 mL/kg/h during more than 24 h, or anuria for more than 12 h [3]. It has been demonstrated that factors such as advanced age, diabetes mellitus, pre-existing renal impairment, intravascular volume depletion after surgery, congestive heart failure, or concurrent use of other nephrotoxic medicines enhance the chance of developing PC-AKI [4,5,6,7,8]. Additionally, sarcopenia—musculoskeletal loss was associated with AKI occurrence after AAA repair—is known to be an independent mortality factor following AAA therapy. According to studies [9,10] PC-AKI is the third most common reason for hospital-acquired acute renal failure and is linked to a high mortality rate. Due to this, a number of methods (such as pre and posthydration and the injection of N-acetylcysteine) have been shown to be somewhat useful in preventing CIN [11], but they are not always practical, particularly when treating patients who have suffered serious injuries. Additionally, there are other potential causes of the creatinine rise in individuals who have had serious injuries (e.g., contrast-induced, haemorrhagic shock, blood transfusions, advanced age). Abdominal aortic aneurysm (AAA) is a degenerative disease of the aortic wall with potentially fatal complications. The widespread adoption of endovascular aneurysm repair (EVAR) for AAA is due to the obvious advantages of the procedure compared to the traditional open repair. However, these advantages have to be weighed against the increased risk of renal dysfunction with EVAR [12,13,14]. Acute kidney injury (AKI) is a severe complication after infrarenal abdominal aortic aneurysm repair [15]. Although acute kidney injury following EVAR is relatively common, its true incidence is still uncertain. The purpose of this study was to assess the incidence of acute kidney injury among patients treated with endovascular repair of ruptured AAA. In addition, we aimed to determine the predictors of PC-AKI in patients with abdominal aortic aneurysm treated with EVAR.

## 2. Results

The data of 192 patients were analyzed. The median age of those patients was 73 years. The proportion of female patients was 24%. The group of 36 patients developed PC-AKI, which is 19% of the entire study population. Mean iodine contrast volume was 149.6 mL in the whole population, 179.3 mL in the PC-AKI population, and 142.1 mL in the non-PC-AKI population. Mean aortic diameter for all study participants was 57.2 mm. Aortic diameter in the population with PC-AKI was 66.9 mm, and in the population without PC-AKI, it was 55.7 mm. Chronic kidney disease pre-existed in 67 of all study participants: 24 patients with PC-AKI and 43 patients without PC-AKI. Table 1 shows that in the male population, pre-existing chronic kidney disease and blood urea nitrogen concentration were significantly higher in the PC-AKI group (*p* < 0.001). In Figure 1, receiver operating curves for aortic aneurysm diameter in predicting postcontrast-induced kidney injury are presented for all patients and for CKD patients.

In Table 2, we compared patients in relation to the aortic aneurysm diameter (cut off ≥67 mm). Hypertension, diabetes, chronic kidney disease, male gender, and incidence of PC-AKI were more prevalent in patients with higher aortic aneurysm diameter ≥67 mm. In addition, in multiple regression analyses independent predictors of PC-AKI were serum creatinine, chronic kidney disease, male gender, and aortic aneurysm diameter ≥67 mm. 

The correlations between eGFR and iodine contrast volume, as well as with aortic aneurysm diameter, determined using Pearson’s correlation, are presented as rank correlation (r) in Figure 2. In Table 3, we assess that PC-AKI is more common in the group with aortic diameter greater than the cut-off point (67 mm) (OR 1.364) with *p* = 0.01.

## 3. Discussion

Nowadays, due to its clear advantages in reducing perioperative morbidity and mortality, EVAR has become the standard procedure for AAA repair [12,13,14]. This study supports previous findings that EVAR has a significant impact on renal function. Together with other published studies that utilized defined criteria to identify AKI, it can be said that 15% to 20% of EVAR patients develop AKI in the immediate postoperative interval. [15,16,17,18,19] Saratzis et al. divided a variety of AKIs after EVAR, which are: microembolization, suprarenal fixation, accessory renal artery blockage, and CIN, as well as the related inflammatory and ischemic reactions [19]. This study corresponds with the analysis of Lee et al. that the main cause of post-EVAR AKI is related to contrast due to association of contrast dose and AKI frequency [20]. In contrast-induced nephropathy, the administration of contrast results in an increase in vasoconstrictive forces, a reduction in local prostaglandin and nitric oxide-mediated vasodilatation, a direct toxic effect of oxygen free radicals on renal tubular cells, an increase in oxygen consumption, an increase in intratubular pressure due to contrast-induced diuresis, an increase in urinary viscosity, and tubular obstruction. As with other studies [15] this analysis confirms that AKI is more common in people with decreased renal function prior to EVAR (baseline). Those who developed AKI in this group had a substantially lower eGFR at baseline. This study does not include perioperative hydration as a prophylaxis of PC-AKI, although proper patient preparation prior to EVAR utilizing hydration with sodium bicarbonates and N-acetylcysteine seems to offer better postoperation outcomes [15]. Due to a lack of data supporting the usage of N-acetylcysteine or hydration with sodium bicarbonates [21], more research is diligently needed to evaluate additional strategies as well as the best strategy to provide hydration to this group of patients. Compared to data analyzed by Krasznai et al. [22] where the incidence of PC-AKI amongst the population with eGFR < 30 mL/min/1.73 m² is more than 50%, in this study 64% of patients with eGFR < 30 mL/min/1.73 m² developed AKI. It is well recognized that contrast media can result in renal vasoconstriction, hypoxia damage, and acute tubular necrosis. [23,24] Contrast volume between patients who developed PC-AKI (179.3cc) versus patients who did not develop AKI (142.1cc) was much less dispersed compared with Mun et al. [25] in whose study it was 249.17cc in PC-AKI patients and 179.43cc in non-PC-AKI patients. Although the Dutch Randomized Endovascular Aneurysm Management (DREAM) trial demonstrated that post Hosmer-Lemeshow operative changes in SCr were comparable with no statistical differences in the incidence of AKI (as per their definition) or need for dialysis, [26] this study significantly demonstrates that adequate patient preparation prior to the procedure as well as management perioperatively aid in the prevention of PC-AKI. Other authors, such as Rastogi et al. [27], estimated that the aneurysmal neck was an important factor in the development of PC-AKI, and the wider it is, the higher the incidence of PC-AKI. This study shows that the aneurysm diameter itself was also connected with the rate of PC-AKI development, and in patients with an aneurysm diameter higher than 67 mm it was 30%, and in those with diameter lower than 67 mm it was 8.1%. In addition, in the recent paper by Arbănași et al. [28] the diameter of the abdominal aorta at different levels has better accuracy and a higher predictive role of rupture than the maximal diameter of AAA.

## 4. Conclusions

One of the major complications after EVAR is acute kidney injury, which is linked to higher death and morbidity rates. This study showed that pre-existing chronic kidney disease is an independent risk factor for postcontrast acute kidney injury. Additionally, male sex and aortic diameter are other independent factors in development of PC-AKI. Only aortic diameter could be a modifiable risk factor, and earlier surgery could be considered to yield better outcomes. More research is critically needed to determine how AKI affects long-term outcomes and to look at preventive options.

## 5. Materials and Methods

This research is a retrospective analysis based on a prospective registry of abdominal aortic aneurysm patients that was conducted at a single center between 2015 and 2021. A group of 247 patients with AAA was operated endovascularly. Due to a lack of clinical data, 55 participants were omitted from this study. In total, data from 192 patients were reviewed for postcontrast acute kidney injury. Additional comorbidities were included in this study: hypertension, diabetes mellitus, atrial fibrillation, chronic coronary syndrome, COPD, and chronic kidney disease. Follow-up examinations were performed before the procedure and 48 h after contrast administration.

### Statistical Analysis

Normality of the distribution was assessed with Shapiro–Wilk test. The characteristics of the studied population are presented as means with standard deviations (SD) for normally distributed continuous, and as the number of cases and percentage (for categorical variables). Statistical significance of differences between two groups were determined using the χ^2^ and Mann–Whitney U tests when appropriate. The correlations between psychical variables and eGFR were determined using Pearson’s correlation. Data are presented as rank correlation (r). The threshold of statistical significance for all tests was set at *p* < 0.05. All analyses were performed using MS Excel (Microsoft, 2020, version 16.40, Redmond, WA, USA) and XL Stat (Addinsoft, 2020, version 2020.03.01, New York, NY, USA). For the purpose of this study, PC-AKI was defined as an absolute increase of serum creatinine ≥0.3 mg/dL or a relative increase ≥150% from baseline value within the first 48–72 h of intervention.

## Figures and Tables

**Figure 1 toxins-15-00130-f001:**
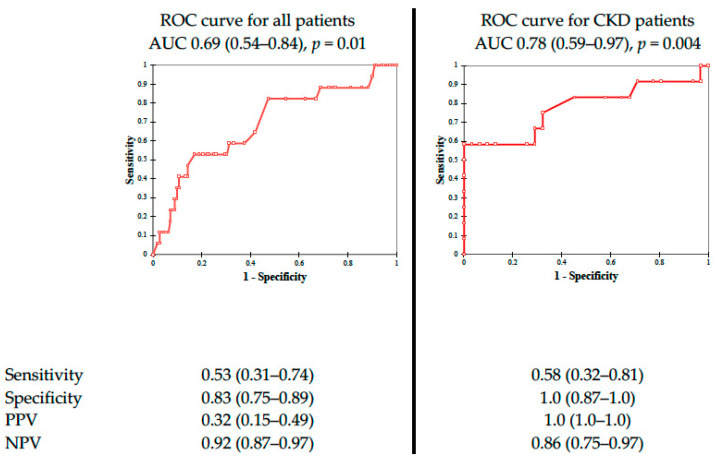
Receiver operating curves for aortic aneurysms diameter in predicting postcontrast- induced kidney injury. Abbreviations: AUC, area under curve, PPV, positive predictive value, NPV, negative predictive value.

**Figure 2 toxins-15-00130-f002:**
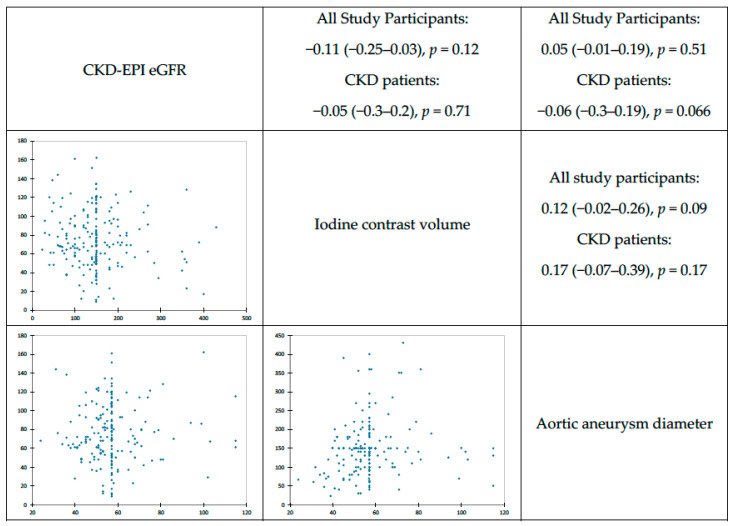
Pearson correlations between physical variables.

**Table 1 toxins-15-00130-t001:** Characteristics of the studied population with comparison of patients with and without postcontrast acute kidney injury.

	All Study Participants (*N* = 192)	Patients with PC–AKI (*N* = 36)	Patients without PC–AKI (*N* = 156)	*p*
Age (years); mean (SD)	73.3 (7.9)	74.9 (9.1)	72.8 (7.5)	0.12
Male; % (N)	76 (146)	86.1 (31)	73.3 (115)	<0.001
Obesity; % (N)	7.8 (15)	11.1 (4)	7.1 (11)	0.42
Hypertension; % (N)	57.8 (111)	55.6 (20)	58.3 (91)	0.76
Diabetes mellitus; % (N)	15.1 (29)	13.9 (5)	15.4 (24)	0.82
Atrial fibrillation; % (N)	17.7 (34)	11.1 (4)	19.2 (30)	0.25
Chronic coronary syndrome; % (N)	32.3 (62)	33.3 (12)	32.1 (50)	0.88
COPD; % (N)	7.3 (14)	0 (0)	9 (14)	0.06
History of acute coronary syndrome; % (N)	13.5 (26)	13.9 (5)	13.5 (21)	0.9
History of stroke; % (N)	9.4 (18)	16.7 (6)	7.7 (12)	0.1
History of neoplasm; % (N)	7.3 (14)	5.6 (2)	11.5 (18)	0.29
Chronic kidney disease; % (N)	34.9 (67)	66.7 (24)	27.6 (43)	<0.001
Serum creatinine concentration (mg/dL); mean (SD)	1.18 (0.72)	1.75 (1.1)	1.1 (0.5)	<0.001
CKD-EPI eGFR mL/min/1.73 m²; mean (SD)	74.77 (30.5)	54.9 (28.7)	79.4 (29)	<0.001
CKD-EPI eGFR <45 mL/min/1.73 m²; % (N)	13.5 (26)	36.1 (13)	8.3 (13)	<0.001
CKD-EPI eGFR <30 mL/min/1.73 m²; % (N)	6.3 (12)	22.2 (8)	2.6 (4)	
CKD-EPI eGFR <15 mL/min/1.73 m²; % (N)	2.6 (5)	8.3 (3)	1.3 (2)	
Blood urea nitrogen concentration (mg/dL); mean (SD)	43.63 (18.9)	56.4 (23.8)	40.6 (16.2)	<0.001
Blood sugar level (mg/dL); mean (SD)	124.5 (49)	141.3 (60.3)	120.7 (45.3)	0.03
Serum sodium concentration (mEq/L); mean (SD)	140.2 (3.3)	139.9 (3.4)	140.3 (3.3)	0.55
Serum potassium concentration (mEq/L); mean (SD)	4.5 (0.5)	4.4 (0.6)	4.5 (0.5)	0.22
Serum chloride concentration (mEq/L); mean (SD)	101.5 (3.8)	101.3 (4.1)	101.6 (3.7)	0.67
Aortic diameter; mean (SD)	57.2 (17)	66.9 (19.7)	55.7 (16)	0.01
Iodine contrast volume (mL); mean (SD)	149.6 (81.2)	179.3 (89.1)	142.1 (77.3)	0.03

Abbreviations: COPD, chronic obstructive pulmonary disease; GFR, estimated glomerular filtration rate; PC-AKI, postcontrast acute kidney injury; SD, standard deviation.

**Table 2 toxins-15-00130-t002:** Comparison of patients with and without aortic aneurysm diameter greater and lower than cut-off point (67 mm).

	Aortic Aneurysms ≥67 mm (*N* = 30)	Aortic Aneurysms <67 mm (*N* = 99)	*p*
Age (years); mean (SD)	76.6 (6.5)	73.2 (7)	0.06
Male; % (N)	86.7 (26)	72.7 (72)	<0.001
Obesity; % (N)	6.7 (2)	4 (4)	0.051
Hypertension; % (N)	56.7 (17)	62.6 (62)	<0.001
Diabetes mellitus; % (N)	10 (3)	18.2 (18)	<0.001
Atrial fibrillation; % (N)	18.7 (5)	18.2 (18)	0.85
Chronic coronary syndrome; % (N)	30 (9)	34.3 (34)	0.66
COPD; % (N)	6.7 (2)	10.1 (10)	0.57
History of acute coronary syndrome; % (N)	13.3 (4)	13.1 (13)	0.98
History of stroke; % (N)	0 (0)	8.1 (8)	0.11
History of neoplasm; % (N)	13.3 (4)	14.1 (14)	0.91
Chronic kidney disease; % (N)	26.7 (8)	35.4 (35)	<0.001
CKD-EPI eGFR <45 mL/min/1.73 m²; % (N)	10 (3)	11.1 (11)	<0.001
CKD-EPI eGFR <30 mL/min/1.73 m²; % (N)	6.7 (2)	5.1 (5)	79.4 (29)
CKD-EPI eGFR <15 mL/min/1.73 m²; % (N)	0 (0)	2 (2)	8.3 (13)
Serum creatinine concentration (mg/dL); mean (SD)	1.11 (0.42)	1.14 (0.54)	0.76
CKD-EPI eGFR mL/min/1.73 m²; mean (SD)	77.3 (30.3)	73.4 (27.7)	0.52
Blood urea nitrogen concentration (mg/dL); mean (SD)	41.9 (17.3)	42.9 (17.4)	0.79
Blood glucose level (mg/dL); mean (SD)	135.5 (46.8)	113.3 (40.6)	0.02
Serum sodium concentration (mEq/L); mean (SD)	139.8 (2.8)	140.7 (3.3)	0.19
Serum potassium concentration (mEq/L); mean (SD)	4.2 (0.3)	4.5 (0.5)	<0.001
Serum chloride concentration (mEq/L); mean (SD)	101.6 (3.3)	101.4 (3.6)	0.83
Iodine contrast volume (mL); mean (SD)	168.1 (99.8)	139.3 (72.1)	0.11
PC–AKI; % (N)	30 (9)	8.1 (8)	<0.001

Abbreviations: COPD, chronic obstructive pulmonary disease; GFR, estimated glomerular filtration rate; PC-AKI, postcontrast acute kidney injury; SD; standard deviation.

**Table 3 toxins-15-00130-t003:** Multivariable logistic regression model. Odds ratio for postcontrast acute kidney injury. The data are presented as odds ratios with 95% confidence intervals (Hosmer–Lemeshow statistic 0.154).

Variables	OR	95% CI	*p*
Serum creatinine concentration	1.439	1.056–1.959	0.02
Age	1.224	0.931–1.608	0.15
Male	1.345	1.004–1.804	0.047
Chronic kidney disease	1.464	1.074–1.995	0.02
Obesity	1.037	0.822–1.307	0.76
Diabetes mellitus	0.951	0.744–1.214	0.69
Hypertension	0.961	0.730–1.265	0.78
Chronic coronary syndrome	1.032	0.756–1.407	0.84
History of acute coronary syndrome	0.945	0.700–1.277	0.71
History of stroke	1.191	0.953–1.489	0.13
History of neoplasm	0.790	0.579–1.079	0.14
Atrial fibrillation	0.765	0.576–1.017	0.07
Aortic aneurysms diameter greater than cut off point (67 mm)	1.364	1.093–1.702	0.01

## Data Availability

The data presented in this study are available on request from the corresponding author.

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
