# Peer review of "Kidney Function, Male Gender, and Aneurysm Diameter Are Predictors of Acute Kidney Injury in Patients with Abdominal Aortic Aneurysms Treated Endovascularly"

_toxins, 2023, doi:10.3390/toxins15020130_

Round 1

Reviewer 1 Report

This is potentially significant research since it provides evidence regarding the kidney function, male gender, and AAA diameter as predictors of AKI risk in AAA following EVAR.

-I suggest authors to include their name and affiliation

-In abstract, please use the same font everywhere

-Authors cand improve the Introduction section with some recent publication regarding the diameter of AAA and nutritional status in AKI risk, as following:

- https://doi.org/10.3390/ijerph192315961

- https://doi.org/10.3390/nu13072212

-At ROC curve, as I see is about the association of AAA diameter with AKI risk in all patients, respectively in AAA patients with CKD. Please correct in Result section line 107-108, and in the Figure 1. caption.

Overall, the manuscript is well written. I want to congratulate the authors.

Author Response

Reviewer 1

This is potentially significant research since it provides evidence regarding the kidney function, male gender, and AAA diameter as predictors of AKI risk in AAA following EVAR.

-I suggest authors to include their name and affiliation It was added accordingly

-In abstract, please use the same font everywhere it was corrected

-Authors cand improve the Introduction section with some recent publication regarding the diameter of AAA and nutritional status in AKI risk, as following: they were cited  

- https://doi.org/10.3390/ijerph192315961 ref 28

- https://doi.org/10.3390/nu13072212 ref 9

-At ROC curve, as I see is about the association of AAA diameter with AKI risk in all patients, respectively in AAA patients with CKD. Please correct in Result section line 107-108, and in the Figure 1. caption.

Table 1 shows that male population, pre-existing chronic kidney disease and blood urea nitrogen concentration were significantly higher in PC-AKI group (P<0.001). On the Figure 1 Receiver operating curves for aortic aneurysms diameter in predicting post contrast induced kidney injury are presented for all patients and for CKD patients.

Reviewer 2 Report

The authors investigate AKI after EVAR in this retrospective study. Congratulations for you effort.

I have two major concerns:

1) did the authors investigate AKI after EVAR or PC-AKI after EVAR. This should be clear throughout the article

2) definition of PC-AKI should have a relevant reference.

3) in my opinion the results should be a little more detailed.

Author Response

The authors investigate AKI after EVAR in this retrospective study. Congratulations for you effort. I have two major concerns:

1) did the authors investigate AKI after EVAR or PC-AKI after EVAR. This should be clear throughout the article It was PC-AKI as the contrast was used  

2) definition of PC-AKI should have a relevant reference.

KDIGO defines PC-AKI as acute kidney injury caused secondary to contrast administration. The definition of AKI provides one of the following: an increase in SCr of ≥0.3 mg/dl (≥26.5 µmol/l) within 48 hours; an increase in SCr of ≥1.5 times baseline that is known or presumed to have occurred within the previous 7 days; or an increase in urine volume of 0.5 ml/kg/h for 6 hours. AKI can be divided into 3 stages. Stage 1 requires one of the following: a reduction in urine production to <0.5 ml/kg/hr during a 6 hour block, increase in SCr by ≥0.3 mg/dl (≥26.5 µmol/l) within 48 hours or increase in SCr to ≥1.5-1.9 times baseline, which is known or presumed to have occurred within the prior 7 days. Stage 2 demands one of the following: serum creatinine increase ≥ 2.0-2.9 times baseline or reduction in urine production to <0.5 ml/kg/hr during two 6 hour blocks. Stage 3 finally is defined by one of the following: serum creatinine increase ≥3.0 times baseline, serum creatinine increase to >4.0mg/dl (353 µmol/l), initiation of renal replacement therapy, reduction in urine production to <0.3 ml/kg/h during more than 24 hours or anuria for more than 12 hours. [3]

3) in my opinion the results should be a little more detailed. It was added as below

Mean aortic diameter for all study participants was 57.2 mm. Aortic diameter in population with PC-AKI was 66.9 mm and in population without PC-AKI was 55.7 mm. Chronic kidney disease pre-existed in 67 of all study participantss, 24 patients with PC-AKI and 43 patients without PC-AKI. Table 1 shows that male population, pre-existing chronic kidney disease and blood urea nitrogen concentration were significantly higher in PC-AKI group (P<0.001). On the Figure 1 Receiver operating curves for aortic aneurysms diameter in predicting post contrast induced kidney injury are presented for all patients and for CKD patients.